# Evaluation of the Links between Lamb Feed Efficiency and Rumen and Plasma Metabolomic Data

**DOI:** 10.3390/metabo12040304

**Published:** 2022-03-29

**Authors:** Florian Touitou, Flavie Tortereau, Lydie Bret, Nathalie Marty-Gasset, Didier Marcon, Annabelle Meynadier

**Affiliations:** 1GenPhySE, Université de Toulouse, INRAE, ENVT, F-31326 Castanet-Tolosan, France; flavie.tortereau@inrae.fr (F.T.); nathalie.marty-gasset@inrae.fr (N.M.-G.); annabelle.meynadier@envt.fr (A.M.); 2Ecole Nationale Vétérinaire de Toulouse, F-31300 Toulouse, France; lydie.bret@envt.fr; 3INRAE, Experimental Unit P3R, F-18390 Osmoy, France; didier.marcon@inrae.fr

**Keywords:** residual feed intake, NMR, plasma, rumen, multivariate analysis

## Abstract

Feed efficiency is one of the keystones that could help make animal production less costly and more environmentally friendly. Residual feed intake (RFI) is a widely used criterion to measure feed efficiency by regressing intake on the main energy sinks. We investigated rumen and plasma metabolomic data on Romane male lambs that had been genetically selected for either feed efficiency (line rfi−) or inefficiency (line rfi+). These investigations were conducted both during the growth phase under a 100% concentrate diet and later on under a mixed diet to identify differential metabolite expression and to link it to biological phenomena that could explain differences in feed efficiency. Nuclear magnetic resonance (NMR) data were analyzed using partial least squares discriminant analysis (PLS-DA), and correlations between metabolites’ relative concentrations were estimated to identify relationships between them. High levels of plasma citrate and malate were associated with genetically efficient animals, while high levels of amino acids such as L-threonine, L-serine, and L-leucine as well as beta-hydroxyisovalerate were associated with genetically inefficient animals under both diets. The two divergent lines could not be discriminated using rumen metabolites. Based on phenotypic residual feed intake (RFI), efficient and inefficient animals were discriminated using plasma metabolites determined under a 100% concentrate diet, but no discrimination was observed with plasma metabolites under a mixed diet or with rumen metabolites regardless of diet. Plasma amino acids, citrate, and malate were the most discriminant metabolites, suggesting that protein turnover and the mitochondrial production of energy could be the main phenomena that differ between efficient and inefficient Romane lambs.

## 1. Introduction

Meeting society’s expectations in terms of environmental preservation, animal health and welfare, and meat quality while staying economically viable is a challenge breeders must overcome. Feeding costs are one of the most important production costs, especially in sheep meat production [1]. Increasing animal feed efficiency is a good way to reconcile economy and ecology, reducing both food inputs and environmental impacts [2,3,4]. The concept of feed efficiency consists of focusing on the relations between what animals eat and what they produce instead of focusing on production only.

Feed efficiency is mainly studied through the feed conversion ratio (FCR) and residual feed intake (RFI). RFI has long been proven heritable in cattle [5] and, more recently, in sheep with estimates ranging from 0.11 to 0.46 [6,7,8]. In the Romane breed, heritability of RFI is estimated to be 0.45 ± 0.08 [9,10]. RFI presents the advantages of being independent (at both the phenotypic and genetic levels) of average daily gain (ADG) [7] compared to FCR and being more suited to genetic selection [11].

The biological determinants of feed efficiency are widely studied in cattle [12], and some studies are emerging in lambs [13,14]. In sheep, feed efficiency has been recently linked to feeding behavior [9,15] and methane emissions [3]. In beef cattle, some studies tried to link RFI with infrared thermometry used as a proxy for radiated heat loss, with conflicting results [16,17]. Methane emissions and radiated heat loss are responsible for a diminution in energy efficiency. To this day, the main hypotheses explaining divergences in feed efficiency rely on differences in digestion and/or metabolism, including processes of protein renewal in skeletal muscle [18]. Indeed, protein synthesis and degradation have an energy cost and have been identified as important factors influencing feed efficiency in beef [19]. Several studies linked rumen metabolites or plasma metabolites with either greenhouse gases emissions [20] or production phenotypes, including RFI [21,22].

Metabolomics studies give an unbiased insight into metabolic phenomena that occur both in the rumen and in the organs and could explain differences in feed efficiency. For example, in the rumen, an orientation of carbohydrate metabolism towards acetate and butyrate production is associated with an increase in methane production by archaea, and hence an increase in methane release and energy loss [20]. Concerning host metabolism, the plasmatic concentrations of amino acids have been associated with differences in protein turnover both in sheep and beef cattle, highlighting the interest of studying those concentrations in divergent animals in terms of feed efficiency [21,22,23].

The above-mentioned studies only focused on the phenotypic RFI, which is the result of the genetic value and environmental factors that are not always well considered because of the limited sizes of experiments. In this paper, our objective was to identify ruminal and/or plasmatic metabolites associated with differences in RFI both at the genetic and phenotypic levels in the progeny of Romane rams divergently selected for RFI under a 100% concentrate diet and belonging to lines called rfi− (efficient animals) and rfi+ (inefficient animals). These differences were identified while lambs were receiving two different diets: a diet rich in concentrates and then a diet rich in forage.

## 2. Results

### 2.1. Zootechnical Parameters

A total of 277 animals (135 from the divergent line rfi− and 142 from the divergent line rfi+) were phenotyped from 12 to 18 weeks of age under a 100% concentrate diet. Their phenotypes are presented in Table 1A. Average Daily Feed Intake (ADFI_C_) was recorded every day, body weight was measured at the beginning and end of the 6-week trial, and ADG_C_ was calculated. Back fat thickness (BFT_C_) and muscle depth (MD_C_) of the *longissimus dorsi* were measured by ultrasound at the end of the trial. Out of the 277 animals, and after calculations of their genetic breeding values, 167 animals with extreme breeding values were kept for the MIX phase, and the same phenotypes were recorded (Table 1B).

#### 2.1.1. Growth Phase under a 100% Concentrate Diet

Descriptions of traits recorded under a 100% concentrate diet (CONC phase) are presented in Table 1A. Animals from rfi+ and rfi– divergent lines ate 2173 and 1991 g/day of concentrate, respectively, these two quantities being significantly different (*p* < 0.001). While marginal, though significant, differences were observed in end-phase metabolic weight (E-MW_C_) and MD_C_ between the two lines, no difference was observed either in ADG_C_ or in BFT_C_. RFI_C_ was obtained from a linear model regressing ADFI_C_ on E-MW_C_, ADG_C_, MD_C_, and BFT_C_ (R^2^ = 0.69, *p* < 2.2 × 10^−16^). E-MW_C_ and ADG_C_ were the most explanatory covariates and neither of the body composition parameters were significant. Mean RFI was found to be negative in line rfi− (−71.4 g/day) and positive in line rfi+ (66.6 g/day), the difference being significant (*p* < 0.001). Genetic and phenotypic RFI_C_ were highly correlated (r = 0.72, *p* < 2.2 × 10^−16^), both being well correlated with ADFI_C_ (r = 0.50, *p* < 2.2 × 10^−16^ and r = 0.55, *p* < 2.2 × 10^−16^, respectively). In the CONC phase, 203 animals out of the 277 (100 from line rfi− and 103 from line rfi+) had phenotypic RFI_C_ matching their genetic line, while 35 lambs from line rfi− had a positive phenotypic RFI_C_ and 39 lambs from line rfi+ had a negative phenotypic RFI_C_ (Table 2).

E-MW_C_ and ADG_C_ were both significantly correlated with ADFI_C_ (r = 0.81, *p* < 2.2 × 10^−16^ and r = 0.49, *p* < 2.2 × 10^−16^, respectively). By construction, RFI_C_ was not correlated to any zootechnical parameters except ADFI_C_.

#### 2.1.2. Finishing Phase under a Mixed Diet

Descriptions of traits recorded under a mixed diet (MIX phase) are presented in Table 1B. Animals from lines rfi+ and rfi− tested in the summer period ate 1817 and 1804 g/day, respectively. Animals from line rfi+ and rfi− tested later in life (i.e., during the fall period) ate 1869 and 1856 g/day, respectively. In 2018, concentrate and forage were distributed as a total mixed ration; therefore, it was not possible to precisely measure consumption of one or the other. In 2019 and 2020, concentrate consumption was limited to 700 g/day, and except for one animal from 2019, all animals ate the whole 700 g daily. ADFI_M_ and average daily forage intake (ADForI_M_) were not significantly different between the divergent lines (Table 1B). No difference was observed between the two lines in terms of E-MW_M_, ADG_M_, and MD_M_, and animals from line rfi− had a slightly higher BFT_M_ than animals from line rfi+. In comparison with the model used to estimate RFI_M_ under a concentrate diet, an additional effect of the period of control nested in the year was added as a fixed effect, and together with zootechnical parameters, it explained 46% of the variation in ADFI_M_ (*p* = 3 × 10^−16^) in the MIX phase. There were no differences between the two lines when RFI_M_ was calculated with total ADFI_M_. For the 2019 and 2020 animals, no difference was observed between the lines when RFI was calculated considering forage intake only (RForI_M_). Out of the 83 animals from line rfi− kept in the MIX phase, only 45 turned out to have a negative phenotypic RFI_M_, and out of the 84 animals from line rfi+, 41 had positive phenotypic RFI_M_ values (Table 3). ADFI_M_ was significantly correlated with E-MW_M_ (r = 0.40, *p* = 6.6 × 10^−7^) and ADG_M_ (r = 0.20, *p* = 0.03).

Since concentrate intake was limited to 700 g/day in 2019 and 2020, animals that ate less forage had a total ration that was denser in energy and crude protein. However, in the Wilcoxon signed-rank tests applied to the RFI_M_ calculated from ADFI_M_, total energy intake or total crude protein intake led to the incapacity of rejecting the null hypothesis, meaning the distributions of the three traits were the same.

### 2.2. Plasma Metabolites

#### 2.2.1. Growth Phase under a 100% Concentrate Diet

Thirty-four metabolites were identified in the plasma of the animals in the CONC phase, the most abundant being D-glucose (Appendix A). All metabolites were included in the PLS-DA model used to discriminate genetic lines rfi+ and rfi−. Based on the area under the receiving operator characteristic curve (AUROC), we retained eight components (AUROC = 0.81) for the PLS-DA of plasma metabolites in the CONC phase (Figure 1A). According to the permutation test, the genetic lines were significantly discriminated by the PLS-DA (*p* = 9.9 × 10^−5^). Only citrate and malate had a variable importance in projection (VIP) value higher than 1.5 in the eight-dimension PLS-DA model and were therefore the two most discriminant metabolites, followed by L-leucine, beta-hydroxyisovalerate, and L-threonine with VIP values around 1.3 (Figure 1B). The loadings of the first component highlighted an association between line rfi− and higher levels of plasma citrate and malate and between line rfi+ and higher levels of L-leucine, L-threonine, and beta-hydroxyisovalerate (Figure 1C).

When PLS-DA was applied to the two extreme phenotypic RFI groups comprising animals with a phenotypic RFI_C_ higher than 0.5 standard deviations (SD) or lower than −0.5 SD independently from their genetic line, the results were similar. Eleven components were retained based on the AUROC (AUROC = 0.84), and they significantly discriminated the two phenotypic RFI_C_ groups (*p* = 1.0 × 10^−3^). When the eleven components were taken into account in the PLS-DA model, only beta-hydroxyisovalerate had a VIP value higher than 1.5, followed by L-glutamate, L-serine, and citrate (VIP = 1.40, 1.29, and 1.25, respectively) (Appendix A).

Pearson correlations were calculated between phenotypes recorded during the CONC phase and the relative concentrations of all metabolites. Only weak correlations were found between metabolites and phenotypes. The highest correlation with ADFI_C_ was with beta-hydroxyisovalerate (r = 0.40, *p* = 8.4 × 10^−11^), which was also significantly correlated with phenotypic RFI_C_ (r = 0.30, 1.7 × 10^−7^). Amino acids were all positively correlated with each other, with the highest correlation being observed between L-leucine and L-valine (r = 0.91, *p* < 2.2 × 10^−16^). Citrate was highly correlated with malate (r = 0.94, *p* < 2.2 × 10^−16^) (Figure 2).

#### 2.2.2. Finishing Phase under a Mixed Diet

Twenty-three metabolites were identified in plasma samples from the MIX phase, 21 of them being present in all samples and D-glucose being once again the most abundant (Appendix A). Based on the AUROC, nine components were retained for the PLS-DA (AUROC = 0.87) (Figure 3A), which significantly discriminated the two divergent genetic lines (*p* = 3.0 × 10^−4^) according to the permutation test. When the nine components were taken into account in the PLS-DA model, only citrate, associated with line rfi−, had a VIP value above 1.5, while L-threonine and L-leucine, associated with line rfi+, had VIP values higher than 1.4 (Figure 3B). The loadings of the first component highlighted the association between line rfi− and citrate and between line rfi+ and L-threonine and L-leucine (Figure 3C).

When PLS-DA was applied to phenotypic RFI groups, the best model in terms of diagnosis had nine components (AUROC = 0.74) but did not significantly discriminate the two phenotypic groups according to the permutation test (*p* = 0.50).

No correlation between either ADFI_M_ or genetic or phenotypic RFI_M_ and plasma metabolites was higher than 0.5 in absolute values. The strongest correlations between phenotypes and metabolites were observed for ADFI_M_ with either L-glycine or L-leucine (r = −0.23, *p* = 0.004 and r = 0.23, *p* = 0.008, respectively). L-leucine was strongly correlated with L-valine (r = 0.85, *p* < 2.2 × 10^−16^) (Figure 4).

A summary of plasma metabolite correlations observed under both diets is presented in Figure 5. All correlations higher than 0.5 in absolute values and significant after Benjamini–Hochberg correction are shown.

### 2.3. Rumen Metabolites

#### 2.3.1. Growth Phase with a 100% Concentrate Diet

Forty-four metabolites were identified in rumen samples from the CONC phase. Thirty of them were quantified in every sample and fourteen were missing from one or several samples (Appendix A).

Based on the AUROC, ten components were retained for the PLS-DA (AUROC = 0.70) that best discriminated the two divergent genetic lines, but this model did not significantly discriminate them according to the permutation test (*p* = 0.79).

The best PLS-DA model that aimed to discriminate extreme phenotypic groups retained 10 components but failed to significantly discriminate efficient from inefficient animals according to the permutation test (*p* = 0.22).

No statistically significant correlation was found between any of the metabolites and ADFI_C_ and RFI_C_. Correlations between rumen metabolites higher than 0.5 in absolute values were, on the other hand, quite numerous. Generally, correlations among amino acids and among volatile fatty acids (VFA) were positive, whereas correlations between amino acids and VFA were negative. As an example, L-alanine was strongly correlated with L-glycine (r = 0.83, *p* < 2.2 × 10^−16^), L-serine (r = 0.73, *p* < 2.2 × 10^−16^), L-leucine (r = 0.81, *p* < 2.2 × 10^−16^), L-isoleucine (r = 0.78, *p* < 2.2 × 10^−16^), L-methionine (r = 0.77, *p* < 2.2 × 10^−16^) and L-phenylalanine (r = 0.73, *p* < 2.2 × 10^−16^) and was negatively correlated with acetate (r = −0.33, *p* = 5.2 × 10^−7^), propionate (r = −0.42, *p* = 2.8 × 10^−11^) and butyrate (r = −0.30, *p* = 2 × 10^−6^). Cadaverine was also highly positively correlated with the amino acids (Figure 2).

#### 2.3.2. Finishing Phase with a Mixed Diet

Nineteen metabolites were identified in rumen samples from the MIX phase. Eighteen of them were quantified in every sample. The last one, citraconate, was not detected in more than half the samples (88 samples out of 164) and was therefore removed from the metabolites considered for the PLS-DA (Appendix A). Neither the PLS-DA on genetic divergent lines (nine components, AUROC = 0.71) nor the PLS-DA on phenotypic groups (five components, AUROC = 0.71) were found to significantly discriminate the groups (*p* = 0.21 and *p* = 0.77, respectively).

No significant correlation was observed between phenotypes and metabolites. All correlations higher than 0.5 in absolute values between metabolites were positive correlations. Acetate and propionate were positively correlated (r = 0.7, *p* < 2.2 × 10^−16^), isovalerate was correlated with ethylmalonate (r = 0.61, *p* = 1.2 × 10^−16^) and 3-methylxanthine (r = 0.90, *p* < 2.2 × 10^−16^), those two being also correlated (r = 0.63, *p* < 2.2 × 10^−16^). Amino acids were correlated with each other and with cadaverine and negatively correlated with VFA (Figure 4).

### 2.4. Correlations between Rumen and Plasma Metabolites

No correlation higher than 0.5 in absolute values was observed between rumen and plasma metabolites, except between rumen dimethylsulfone and plasma dimethylsulfone in both phases: r = 0.58 (*p* < 2.2 × 10^−16^) and r = 0.70 (*p* < 2.2 × 10^−16^) in the CONC (Figure 2) and MIX (Figure 4) phases, respectively.

## 3. Discussion

### 3.1. Phenotypes

Genetic selection of the divergent lines on RFI was initiated in 2015 from 12 extreme sires phenotyped in 2013 and 2014 [10]. These lines rely on RFI phenotyped in young lambs fed a rich diet. In the first generation of selection [10], animals from the two lines essentially differed in ADFI and phenotypic RFI and showed no statistically significant differences in ADG, E-MW, BFT, or MD. Animals tested in this work were from the second (103 lambs from 2018) and third generations (101 lambs from 2019 and 73 from 2020) of selection. They exhibited a similar pattern, differing in ADFI_C_ and RFI_C_, only marginally in E-MW_C_ and MD_C_, and not at all in ADG_C_ or BFT_C_ under the same concentrate-based diet that was used for selection. One purpose of this study was to evaluate phenotypes in both divergent lines under a mixed diet, implying less competition with human food. The number of animals kept in the MIX phase was lower, but these animals were extreme males in terms of genetic RFI_C_. Despite this, under the MIX diet, no difference between lines was significant with any of the phenotypes considered, even though mean phenotypic RFI_M_ in lines rfi− and rfi+ were still negative and still positive, respectively. This absence of significant differences during the MIX phase could be due to two parameters evolving concomitantly in our study: the diet and the animals’ age and growth status. Redden et al. found that Targhee ewes that were efficient in terms of RFI under a high-energy diet while growing were not the same as the efficient ones when they were fed a forage-rich maintenance diet as yearlings [24]. Oliveira et al. reported similar results in Nellore cattle: animals from the low-RFI group, when fed a high-energy diet in the feedlot, had a 9.4% and 19.7% lower DMI than animals from the medium-RFI and high-RFI groups, respectively, but when animals were brought to the pasture, no significant difference was observed between the groups [25]. In their study, animals were older when switched to grass. The main difference between these studies and ours is that their RFI groups were made up based on single phenotypic assessments while our divergent lines stem from a multiple-generation genetic selection, making classification more reliable. Our selection seems to be appropriate for rapidly growing lamb breeding often based on high-concentrate diets and slaughtering between 3.5 and 6 months of age. Some of our lambs had null or negative mean ADG_M_ during the second phase of measurements, indicating that they were not growing anymore and that potential mechanisms associated with growth, such as preferential muscle deposition over fat [26], could not be applicable anymore on lambs older than 6 months. If this was the sole explanation, we would expect plasma metabolites to differ between efficient and inefficient animals in the CONC phase but not necessarily in the MIX phase, and no difference in rumen metabolites in either phase. The difference between our phases could also be linked with a better use of the concentrate diet in rfi− lambs than in the rfi+ lambs and no better use of the mixed diet. In this scenario, we would expect both rumen and plasma metabolites to vary between lines during the CONC phase as being linked to one or the other line as a result of a different digestion and absorption of this diet in divergent animals [27].

### 3.2. Plasma Metabolites

Few studies have been performed on sheep plasma metabolites, and even fewer have linked NMR-measured metabolites and feed efficiency. In a recent study, Goldansaz et al. used multiple platform analysis to identify potential biomarkers of RFI in sheep [21]. A few months later, Foroutan et al. tried to extensively describe the metabolome of beef cattle with either high-RFI or low-RFI phenotypes using NMR, GC/MS, and LC/MS techniques [28] and compared their results with pre-existing studies on cattle and with scarce studies that exist on sheep [23,29,30].

#### 3.2.1. Amino Acids

Repeatedly throughout the studies, amino acids were more abundant in the plasma or serum of less efficient animals than in the plasma or serum of the more efficient ones. In particular, L-threonine, identified as the most discriminant metabolite for the line rfi+ in our study, was also more represented in high-RFI animals compared to low-RFI ones in other studies [23,28]. This result can be nuanced by the comparison between the two PLS-DAs performed on the genetic lines or the extreme phenotypic RFI groups since, in the latter, plasma L-threonine was not identified as one of the most discriminant metabolites in either of the phases. L-glycine is more controversial since it has been found to be higher in efficient animals in some studies [12,28] and in less efficient animals in others [12,22], and was associated with line rfi+ in the PLS-DA discriminating our divergent lines and associated with efficient animals when phenotypes were considered. L-glutamate, L-serine, and even more beta-hydroxyisovalerate (which is a product of L-leucine metabolism) were the most discriminant metabolites associated with inefficient animals in terms of phenotypic RFI_C_ in the CONC phase. L-glutamate had also been associated with inefficiency in Foroutan et al., 2020 [28]. Beta-hydroxyisovalerate has not been reported yet in the literature either way.

The overall increase in amino acids plasma concentrations could be linked to an increase in crude protein consumption, proportional to the increase in ADFI. However, no correlation was found between any individual amino acid concentration and ADFI, regardless of diet. This may be due to the fact that plasma amino acid concentrations are a reflection of both ingestion and metabolism and that ADFI is a reflection of feed intake during the whole period, not only during the few days or hours prior to sampling. Mechanisms other than simply ingestion must be at stake. Jorge-Smeding et al. suggested that efficient animals could have a slowed-down urea cycle, resulting in the accumulation of carbamoyl phosphate and fumarate and a decrease in plasma concentrations of L-ornithine, L-aspartate, L-valine, and L-lysine [29]. In our case, neither L-ornithine nor L-lysine were quantified in the plasma, but L-citrulline, which is part of the urea cycle, was not differentially expressed in the different genetic lines. However, L-valine had a VIP value higher than 1 in the MIX phase and was associated with line rfi+. Urea is not one of the 191 metabolites in the ASICS library [31] and therefore was not identified in our study; thus, we cannot properly conclude on the acceleration or slow-down of the urea cycle.

Another explanation for feed efficiency, particularly discussed in Cantalapiedra-Hijar et al., is a modification of protein turnover [18]. Protein turnover is the continuous process of the synthesis and degradation of proteins in the organism allowing animals to adapt to their physiological status [32]. Its increase, resulting in a higher energy expense to deposit the same amount of muscle protein, could explain why some animals are poorly efficient. In our study, we found no difference in *longissimus dorsi* depth measured with ultrasounds between efficient and inefficient animals in either phase, suggesting that while eating less, efficient animals deposited the same amount of muscle. In some non-ruminant species, protein turnover, which is known to be variable between individuals, has been proven to be heritable and could therefore be one of the mechanisms involved in feed efficiency [33,34]. Protein turnover is an energy-consuming process that increases with protein concentration of the diet, and both synthesis and degradation are enhanced when supplementation of an amino acid-lacking diet is realized [35]. Our animals were fed in order to cover or overpass their protein needs in accordance with INRAE recommendations and thus cannot be considered as lacking amino acids [36].

#### 3.2.2. Organic Acids

Citrate was the most discriminant metabolite between the two genetic lines in both our phases. It has been identified in most of the studies previously quoted, but its association with efficiency was not commonly reported. Even in our study, when discrimination was made on the bases of phenotypic groups, citrate importance in the discrimination was far lower. Karisa et al. even linked citrate and less efficient animals [23].

Plasma citrate concentrations are mainly regulated by digestive absorption of feed citrate, bone metabolism, and renal filtration and resorption [37,38]. Thus, an increase in citrate plasma concentration in genetically efficient animals could stem from their better reabsorption of urine citrate, resulting in less waste of potentially valuable carbon or higher export from the cell when energy is in excess, resulting in a more efficient use of citrate [37]. Other hypotheses would have been a higher digestive citrate absorption, but this is not consistent with the fact that efficient animals eat less and that the ruminal metabolites profile does not suggest a higher citrate production in the rumen.

Finally, the increase in citrate plasma concentration could be caused by an increased citrate production in the cell or an increased export of citrate from the cell to the plasma. The concomitant augmentation of citrate and malate, also highly discriminant in the CONC phase of our experiment, suggests that when fed 100% concentrate, catabolic phenomena and energy metabolism could be involved in the difference in feed efficiency between lambs (Figure 6). This was previously considered by Herd and Richardson in cattle [19]. Citrate is a key metabolite of both anabolism and catabolism, being the initial step of the citric cycle in the mitochondria and thus a very important element of ATP production in the cell. Citrate is also involved in fatty acid synthesis in the cytoplasm through acetyl-CoA production [39,40] and in modulating glycolysis, lipogenesis, and neoglucogenesis.

Kelly et al. reported an increase in 3-hydroxybutyrate in inefficient heifers [41]. In ruminants, 3-hydroxybutyrate has two potential origins. On one hand, absorbed butyrate can be converted to 3-hydroxybutyrate in the rumen epithelium, and thus, an increase in feed intake or an orientation towards butyrate production can be responsible for a plasma 3-hydroxybutyrate increase [42]. On the other hand, it could result from an impaired energy valorization. When intermediates of the citric cycle are lacking, especially when oxaloacetate is oriented towards neoglucogenesis, citrate synthase is inhibited, resulting in a condensation of acetylCoA into ketone bodies in the liver (Figure 6).

#### 3.2.3. Integration

Another hypothesis could explain both the increased citric cycle intermediates in efficient animals and increased amino acids in inefficient animals (Figure 6). Efficient animals could have a higher capacity to orient excess amino acids towards anaplerotic reactions, resulting in the augmentation of the citric cycle metabolites, including citrate and malate.

Increased citrate and malate mitochondrial concentrations that could stem from an increased deamination of excess amino acids may be better exported from the mitochondria in efficient animals, which would avoid impairing citric cycle reactions, as suggested in MacDonald et al. [43]. Even though plasma exports from the cell to the plasma are not well described for now, contrarily to exchanges between the mitochondria and the cytoplasm [44], Mycielska and coworkers postulated the existence of a transporter allowing citrate and potentially other citric cycle intermediates to leave the cell [45].

The link between mitochondrial metabolism and feed efficiency has been proposed in the literature. Sharifabadi and coworkers isolated mitochondria from lambs differing in ADFI and RFI and found that the activity of all five mitochondrial respiratory chain complexes were enhanced in feed-efficient animals [46]. Moreover, studies on isolated pig mitochondria have shown that pigs selected for efficiency tend to have a lower electron leakage from several complexes of the respiratory chain and a lower production of reactive oxygen species (ROS), correlated with their phenotypic RFI [47]. Similarly, Bottje et al. reviewed links between electron leaks and feed efficiency, concluding that poorly efficient animals had a less efficient electron transfer along the electron transfer chain, leading to an increase in ROS in broilers [48]. In Kolath et al., feed efficiency was associated with greater efficiency in respiratory control ratio, implying a higher degree of coupling between respiration and oxidative phosphorylation but no increase in production of ROS when expressed as a function of respiration rate [49].

A description of the main metabolic pathways highlighted in this study is proposed in Figure 6.

### 3.3. Rumen Metabolites

No significant correlation higher than 0.4 in absolute values was observed between rumen and plasma metabolites, with the notable exception of the correlation between rumen dimethylsulfone and plasma dimethylsulfone in both phases. Plasma dimethylsulfone results from the absorption of rumen dimethylsulfone, presumably produced from the degradation of organic sulfur compounds such as L-methionine or from S-methyl cysteine sulfoxide that would come from the rapeseed meal incorporated in the concentrates. An increase in plasma dimethylsulfone was previously described to be linked to reduced methane emissions in dairy cows, but this was not associated with a production increase [50]. In our study, dimethylsulfone was not linked with any of the phenotypes, nor with genetic RFI.

Rumen metabolites could potentially be linked to feed efficiency in two ways. On one hand, metabolites produced by the rumen microbiota are subsequently released in the rumen and absorbed, and differences in metabolites in the rumen could indicate impaired or enhanced digestion associated with better use of feed. On the other hand, metabolites absorbed in the rumen or in the intestine could influence host phenotype by orienting metabolism towards tissue deposition or by increasing protein turnover, for example. Metabolites that we identified in the rumen are very close to those obtained in earlier studies led with NMR techniques [20]. In our study, rumen metabolites did not discriminate genetic lines nor phenotypic groups, regardless of the diet. Indeed, discrimination by PLS-DA was poor (as highlighted by the AUROC values being 0.7 at most, and *p*-values of the permutation tests being greater than 0.05).

Succinate, which is a microbial intermediate towards propionate production in the rumen [51], was found to be linked to animals growing faster for the same feed intake in a study from Clemmons et al. [52].

Artegoitia and coworkers compared rumen metabolomics of steers with high ADG and steers with low ADG for the same amount of feed ingested [53]. They reported an increase in phenylpyruvate, pyroglutamate, cortisol, DHEA sulfate, lactate, imidazole, and malonyl-carnitine and a decrease in lactate, taurine, and both alpha-linolenic acid and linoleic acid in rapidly growing animals (*p* < 0.1), suggesting that ruminal metabolism may differ between efficient and inefficient animals [53]. Among these metabolites, only pyroglutamate and lactate in the CONC phase and sebacate in the MIX phase were quantified in our study, and neither was associated with either lines or phenotypic RFI.

Two recent studies compared VFA relative concentrations and total VFA concentrations without finding any difference between efficient and inefficient animals [54,55]. Consistently with these studies, no difference was found in the relative concentrations of VFA in our study.

Ruminal digestion and lipid biohydrogenation in the rumen are mainly driven by the microbiota, and investigating ruminal microbiota could highlight differences that are difficult to determine using metabolomics. As an example, Artegoitia et al. also found higher bacterial and lower archaeal populations in efficient animals, and other studies have investigated links between ruminal microbiota and feed efficiency [56,57,58]. The analysis of microbial populations and long-chain fatty acids in our samples might unravel new explanations.

## 4. Materials and Methods

### 4.1. Animals

A total of 277 Romane male lambs belonging to the 2nd and 3rd generations of divergent selection on RFI were bred at the INRAE experimental unit P3R in Bourges (https://doi.org/10.15454/1.5483259352597417E12, 1 February 2022) during the years 2018 (2nd generation), 2019, and 2020 (3rd generation). The genetic selection of the lines has been presented in Tortereau et al. [10]. Briefly, every year since 2015, males were phenotyped for 6 weeks from week 12 to week 18 of age under a 100% concentrate diet, and their RFI values were calculated. Breeding values for RFI were then estimated, and males with the most extreme breeding values for RFI were selected as the sires of the next generation [10]. Lambs inherited their line (rfi− or rfi+ for the efficient or less efficient lines, respectively) from their sire.

In 2018, 2019, and 2020, animals following this phenotyping protocol were bred in 6, 5, and 4 different pens, respectively, each pen being equipped with one automatic concentrate feeder (CONC phase). Lambs were grouped based on their body weights recorded at the beginning of the adaptation period (i.e., when lambs were about 10 weeks of age). After the estimation of the RFI breeding values, the most extreme animals in terms of genetic RFI values were fed a mixed ration distributed by automatic forage and concentrate feeders (MIX phase). This second phase was led between weeks 24 and 30 of age in the summer (29 animals in 2018, 36 in 2019) or between weeks 32 and 38 of age in the fall (29 animals in 2018, 35 in 2019, 38 in 2020) due to the limited number of automatic forage feeders available (Figure 7). In total, up to 30 animals were kept per period in 2018, resulting in keeping 57% of the animals, and up to 40 animals were kept per period in 2019 and 2020, resulting in 69% and 52% of the animals being kept, respectively.

During the MIX phase, animals were divided into two pens per period. In 2018, each pen was equipped with two forage feeders, each delivering the total mixed diet comprising one-third concentrates and two-thirds hay. The total mixed ration was calculated to enable a growth of 125 g/day according to INRA standards [36]. In 2019 and 2020, each pen was equipped with two forage feeders delivering only forage ad libitum, and one concentrate feeder delivering at most 700 g of concentrate per day per animal. This amount was chosen to match the consumption of concentrates in the MIX phase in 2018 and represents approximately one-third of the ration. As a result, the mean proportions of concentrate in the MIX rations in 2019 and 2020 were 34% and 37%, respectively. Animals were housed on litter chips to avoid straw consumption that would distort the measured feed intake.

### 4.2. Diets

In the CONC phase, the diet consisted of a 100% concentrate meal made up of beet pulp, wheat bran, barley, corn, rapeseed meal, sunflower meal, and pelleted lucerne (880 gDM/kg, 18.2% crude protein, 10.5% crude fiber, and 2.7% crude fat in 2018 and 0.3% crude fat in 2019, on a DM basis) fed through automatic feeders. During the MIX phase, the diet consisted of a mixed ration of approximately two-thirds orchard hay (915 gDM/kg, 6.6% crude protein for 2018 and fall 2019 and 8.6% crude protein for 2019; 37% crude fiber in 2018 and 34% crude fiber in 2019) and one-third of a concentrate made of wheat, barley, wheat bran, and rapeseed meal (883 gDM/kg, 20% crude protein, 11% crude fiber, 4% crude fat on a DM basis). Chemical composition of feed was determined according to the procedures of the Association of Official Analytical Chemists (Association of Official Analytical Chemists, 1998) [59]; neutral detergent fiber was determined according to the procedure of Van Soest et al. (1991) [60].

### 4.3. Traits

Animals were weighed at the beginning (B-W) and the end (E-W) of each period of testing. End metabolic weight (E-MW) was calculated as E-W^0.75^. The difference between B-W and E-W was used to calculate the ADG of each animal during the period of testing. MD of *longissimus dorsi* and BFT were both measured by ultrasound at the end of each period of testing. ADFI was determined as the mean of daily cumulative feed intake during the six-week period. Phenotypes from the CONC phase are presented with a C subscript and phenotypes from the MIX phase with an M subscript. Residual feed intake was estimated for each diet separately using the following multiple linear regression for phase CONC:ADFI_C,i_ = µ + β1(E-W_C,i_)^0.75^ + β_2_ADG_C,i_ + β_3_MD_C,i_ + β_4_BFT_C,i_ + RFI_C,i_

The period of control effect nested in each year was added as a fixed effect in the MIX phase. In this model, metabolic weight is a proxy for maintenance requirements while the other parameters are linked to meat production. To deal with the issue of the energy density of the ration when animals were restricted to 700 g/day of concentrate and allowed ad libitum consumption of a less-energetically dense forage, Wilcoxon signed-rank tests were performed, comparing the residuals of the above-presented model when the predicted variable was ADFI, energy total intake, or crude protein total intake.

### 4.4. Sampling

The 277 animals were sampled during the CONC phase (103 in 2018, 101 in 2019, and 73 in 2020), and 167 were then sampled during the MIX phase (58 in 2018, 70 in 2019, and 37 in 2020). In practice, feeders were stopped the night before the sampling, resulting in a minimum of ten to eleven hours of fasting, and water was stopped 2 h prior to the sampling to avoid an excessive dilution of ruminal content. However, in the CONC phase of 2019 and the fall period of the MIX phase of 2018, the automatic feeders were not stopped (Figure 7).

Rumen fluid was sampled using an esophageal probe and a pump, transferred to 2 mL microtubes and immediately frozen into liquid nitrogen. Five microtubes containing rumen samples were broken during the freezing process and one did not have enough rumen fluid for the protocol. Therefore, three samples from the CONC phase and three from the MIX phase were not available for further analysis.

Blood was sampled at the jugular vein using vacutainer needles and tubes containing lithium heparin and centrifuged at 2400× *g* for 10 min at 4 °C. Plasma was transferred to 2 mL microtubes and immediately frozen in liquid nitrogen. All samples were transferred from liquid nitrogen to −80 °C freezers where they were stored. Two samples, one from the CONC phase and one from the MIX phase, were broken during the process and therefore not available for subsequent analysis.

### 4.5. NMR Analysis

#### 4.5.1. Chemical Analysis

Plasma samples both from 2018 and 2019 were prepared at the same time; similarly, rumen fluid samples of these two years were prepared at once. Plasma and rumen samples from 2020 were prepared at the same time. Plasma samples were centrifuged at 3000× *g* for 5 min at 4 °C. Then, 200 µL of the supernatant was transferred to a microtube containing 500 µL of phosphate buffer (pH = 7.0) with 17.2 mg/mL of trimethylsilylpropanoic acid (TSP) used as a reference for chemical shift. This mix was then centrifuged for 15 min at 4190 g and 4 °C. Finally, 600 µL of the supernatant was transferred to NMR tubes. Rumen fluid samples followed the same protocol until the last centrifugation at 4190 g. In this step, the supernatant was transferred to another microtube and centrifuged again in the same conditions, and finally, 600 µL of the supernatant was transferred to NMR tubes. All NMR experiments were performed using the MetaToul-AXIOM platform, a partner of the National Infrastructure of Metabolomics and Fluxomics: MetaboHUB (MetaboHUB-ANR-11-INBS-0010, 2011). Samples were kept at 300 K while spectra were acquired using the cpmgpr1D Bruker pulse program on a Bruker AVANCE III HD 600 MHz NMR spectrometer (Bruker Biospin; Rheinstetten, Germany) operating at 600.13 MHz for ^1^H resonance frequency using an inverse detection 5 mm ^1^H-^13^C-^15^N-^31^P cryoprobe attached to a CryoPlatform (the preamplifier cooling unit) with the following parameters: 512 transient and 16 “dummy” scans using a relaxation time of 2.0 s and an acquisition time of 1.36 s, resulting in the acquisition of 32,000 data points. Plasma and rumen samples were analyzed in two separate batches. Pre-processing of the spectra (zero order phase correction, baseline correction, and shift referencing) was performed using the TopSpin^®^ software (version 4.0.9) from Bruker (Billerica, MA, United States of America). Shift referencing was calibrated on TSP in rumen spectra and on D-glucose doublet at 5.24 ppm for plasma samples because complexation between proteins and TSP made it unavailable for referencing.

#### 4.5.2. Bioinformatic Analysis

Pre-processed spectra were finally analyzed with the ASICS (version 2.5.3) R package, which removed the solvent (water) signals (between 4.5 and 5.1 ppm), normalized areas under the spectra to constant sum, and then identified and quantified metabolites [61]. Each diet × biological fluid (plasma or rumen) combination was treated in a separate analysis. The parameters were the following: noise.thres was set to 0.015 in accordance with noise baseline observed in the spectra, max.shift was set to 0.01 in all spectra, and clean.thres was set to 50 to keep only the metabolites that were present in at least 50% of the samples in each diet × biological fluid combination.

### 4.6. Statistical Analyses

Production traits and metabolite quantifications were corrected using the following linear models:CONC phase: Y~µ + Year + Pen%in%Year + Fasting + ε(1)
MIX phase: Y~µ + Year + Period%in%Year + Pen%in%Period%in%Year + Fasting + ε(2)

The fasting fixed effect (2 levels) was introduced to take into account that in one of the three years of the CONC phase and one of the five periods of the MIX phase, feeders were not stopped prior to sampling (Figure 7).

Principal component analysis was performed on each metabolomic dataset to identify potential outliers, resulting in the exclusion of an animal from the CONC phase that had atypical plasma and rumen profiles and had barely eaten the two days prior to the sampling. As a result, 274 and 273 samples were available for subsequent analyses on plasma and rumen metabolites, respectively, during the CONC phase, and 166 and 164 samples were available for plasma and rumen analysis, respectively, during the MIX phase (Appendix A).

Partial least square discriminant analysis (PLS-DA) from the mixOmics R package was applied on residuals obtained from the linear models (1) and (2) to examine the links between genetic lines and metabolites in rumen or in plasma under each diet [62]. Area under the receiver operating characteristic curve (AUROC) was used to select the number of components to consider in the discriminant analyses. This method relies on the probability of correctly classifying a sample in a binary classification system compared to the probability of wrongly classifying the sample. A perfect classifier would always correctly classify and therefore have an AUROC of 1. An AUROC of 0.5 characterizes a random classifier. Validation of the discrimination performance was assessed by the MVA.test function from the RVAideMemoire R package, with 10,000 permutations [63]. Then, loadings on the first component, being the most discriminant, and variable importance in projection (VIP) with the optimal number of components according to AUROC taken into account were used to determine which variables were the most explanatory.

Additional PLS-DAs were plotted using phenotypic RFI as groups. The groups were formed by excluding individuals whose phenotypic RFI was between mean −0.5 SD and mean +0.5 SD. Animals with phenotypic RFI higher than 0.5 SD were considered efficient at the phenotypic level and animals with phenotypic RFI lower than 0.5 SD were considered inefficient at the phenotypic level. These PLS-DAs allowed us to compare genetic RFI and phenotypic RFI in both phases in order to determine if the differences potentially identified between the divergent lines were consistent with the phenotypic evaluation.

Pearson correlation coefficients between metabolites and phenotypes were calculated within each diet × biological fluid combination, and the Benjamini–Hochberg correction was applied to the correlations [64]. Pearson correlations between plasma and rumen metabolites within each diet were also calculated and submitted to Benjamini–Hochberg correction.

## Figures and Tables

**Figure 1 metabolites-12-00304-f001:**
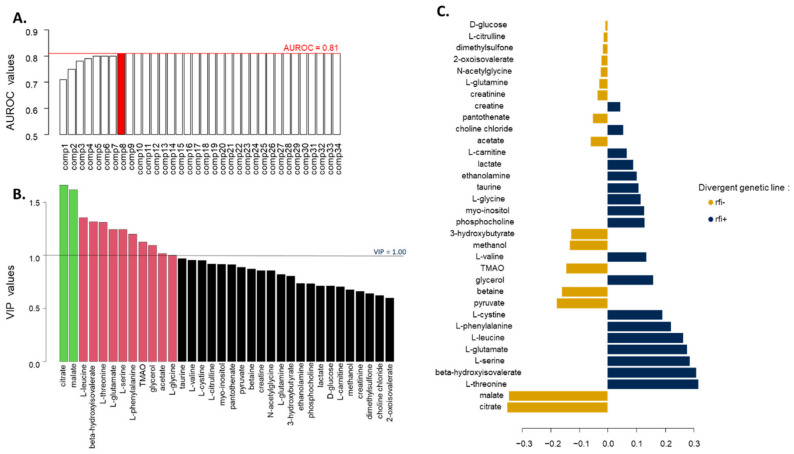
Results from the PLS-DA of the plasma metabolites measured during the CONC phase according to the rfi divergent genetic line of lambs (n = 133 rfi− and n = 142 rfi+). (**A**) Accuracy of the PLS-DA models assessed using AUROC; red line marks the maximum AUROC value obtained with 8 components. (**B**) Selection of the variables contributing the most to the discriminant analysis using a VIP approach. (**C**) Loading values of each metabolite on the first component of the PLS-DA model; in gold and blue, metabolites associated with line rfi− and line rfi+, respectively.

**Figure 2 metabolites-12-00304-f002:**
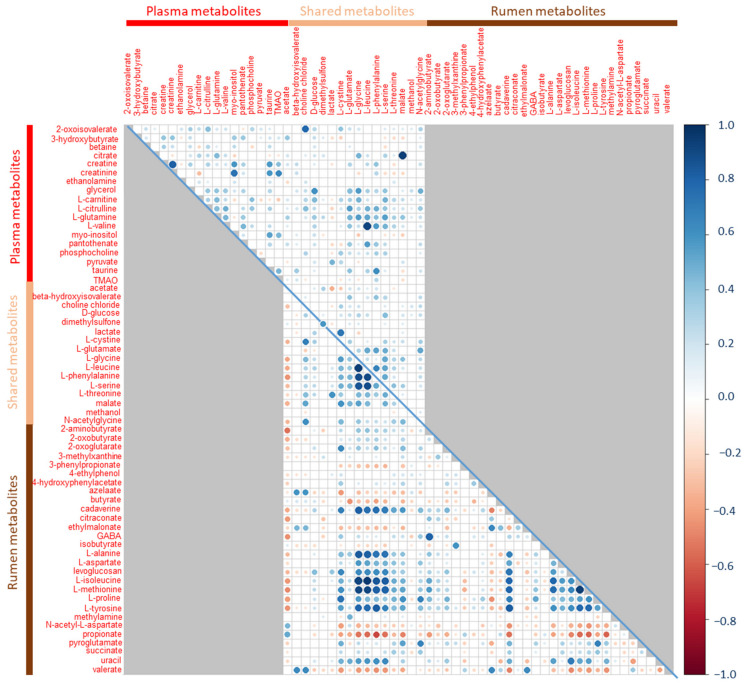
Correlation matrix of metabolites measured during the CONC phase in plasma (above the diagonal) and rumen (below the diagonal). Correlations of a metabolite between the two biological fluids are given on the diagonal. Gray squares correspond to non-existing correlations. White squares are non-significant correlations after Benjamini–Hochberg adjustment of *p*-values.

**Figure 3 metabolites-12-00304-f003:**
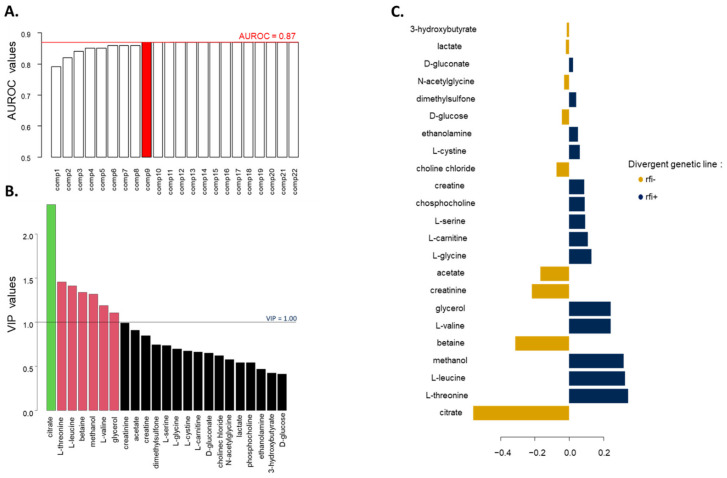
Results of the PLS-DA of the plasma metabolites measured during the MIX phase according to the rfi divergent genetic line of lambs (n = 133 rfi− and n = 142 rfi+). (**A**) Accuracy of the PLS-DA models assessed using AUROC; the red line marks the maximum AUROC value obtained with nine components. (**B**) Selection of the variables contributing the most to the discriminant analysis using a VIP approach. (**C**) Loading values assigned to each metabolite on the first component of the PLS-DA model; in gold and blue, metabolites associated with line rfi− and line rfi+, respectively.

**Figure 4 metabolites-12-00304-f004:**
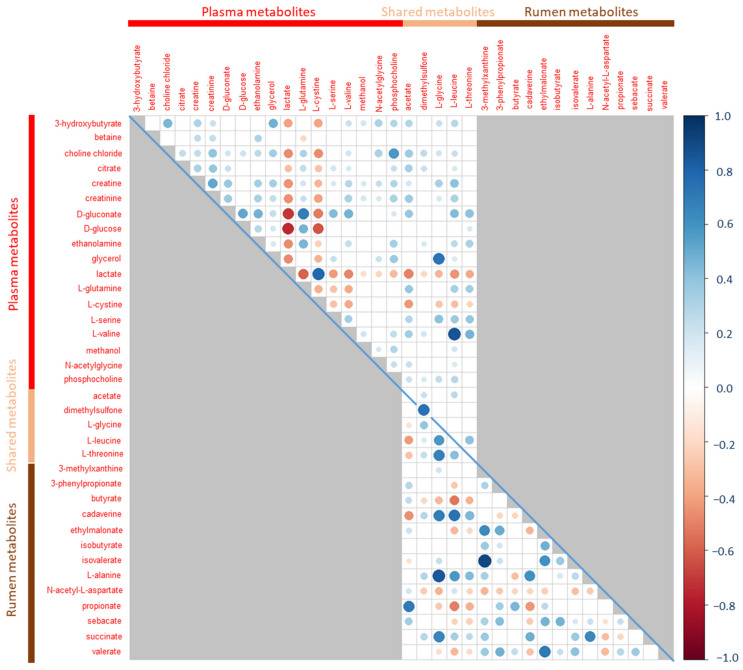
Correlation matrix of metabolites measured during the MIX phase in plasma (above the diagonal) and rumen (below the diagonal). Correlations of a metabolite between the two biological fluids are given on the diagonal. Gray squares correspond to non-existing correlations. White squares are non-significant correlations after Benjamini–Hochberg adjustment of *p*-values.

**Figure 5 metabolites-12-00304-f005:**
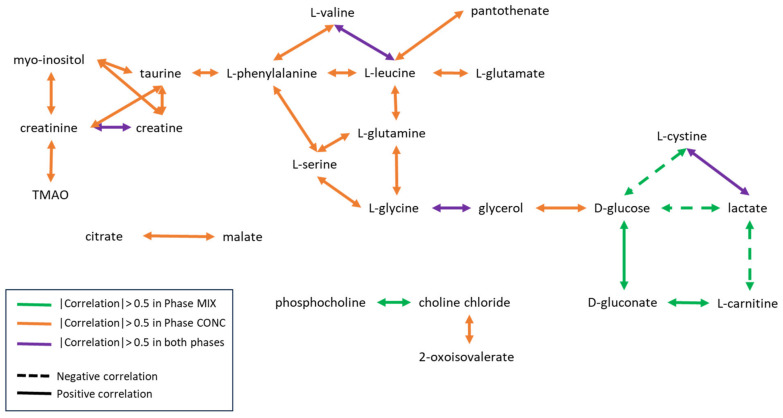
Summary of plasma metabolite correlations. Significant correlations higher than 0.5 in absolute values specifically in the CONC phase, the MIX phase, or under both diets are represented in orange, green, and purple, respectively. Negative correlations are marked by dashed lines and positive correlations are marked by solid lines.

**Figure 6 metabolites-12-00304-f006:**
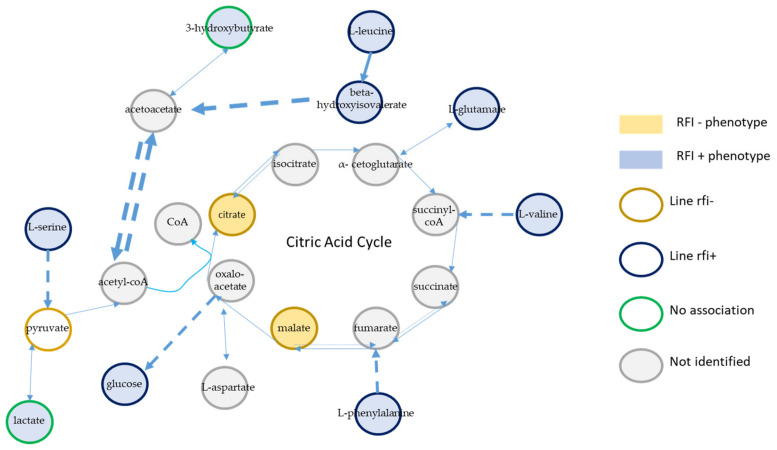
Summary of metabolites found to be associated with genetic lines or phenotypic RFI groups. Gold circles and blue circles indicate that metabolites were associated with line rfi− and line rfi+, respectively. Gray circles indicate that the metabolites were not identified in our study. Green circles indicate that the metabolites were identified but not significantly associated with either of the lines. Gold and blue backgrounds represent the association between a metabolite and rfi− or rfi+ phenotypic groups, respectively. Dashed lines represent several reactions in a pathway. Dotted lines represent potentially reversible reactions.

**Figure 7 metabolites-12-00304-f007:**
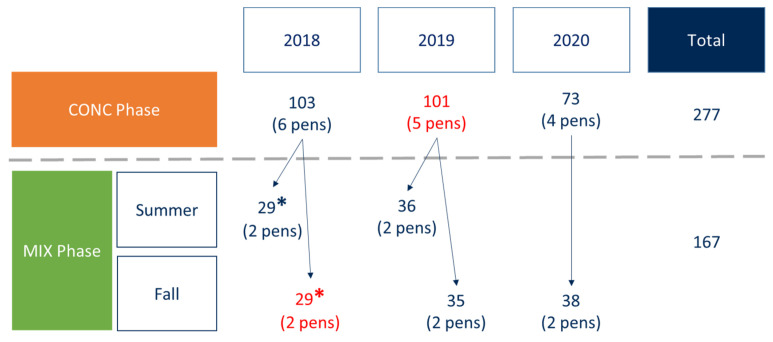
Experimental design. Periods marked in red indicate that the feeders were not stopped prior to sampling. * Concentrate and forage distributed as a TMR through forage feeders only.

**Table 1 metabolites-12-00304-t001:** Descriptive statistics of phenotypes during CONC (A) and MIX (B) phases, and estimation of the divergent line effect.

**A**	**Phase CONC (N = 277)**							
**Traits ***	**Mean (SD)**	**rfi−**		**rfi+**		** *p* ** **-Value**		
ADFI_C_ ^1^ (g/day)	2098 (273)	1991		2173		5 × 10^−12^		
E-MW_C_ ^2^ (kg)	20.6 (1.9)	20.3		20.7		0.0014		
ADG_C_ ^3^ (g/day)	327.6 (62.4)	326.5		326.5		0.93		
BFT_C_ ^4^ (mm)	5.73 (0.87)	5.81		5.74		0.41		
MD_C_ ^5^ (mm)	28.1 (2.4)	27.8		28.3		0.02		
RFI_C_ ^6^ (g/day)	0 (151.4)	−71.4		66.6		<2.2 × 10^−16^		
**B**	**Phase MIX (N = 167)**							
**Traits ***	**Mean (SD)**	**rfi−**		**rfi+**		**rfi−**	**rfi+**	** *p* ** **-Value**
		Summer	Fall	Summer	Fall			
ADFI_M_ ^1^ (g/day)	1857 (280)	1804	1856	1817	1869	1830	1843	0.80
E-MW_M_ ^2^ (kg)	22.8 (1.5)	22.7	22.6	22.9	22.8	22.7	22.8	0.38
ADG_M_ ^3^ (g/day)	124.4 (64.3)	147	113	137	104	130	121	0.40
BFT_M_ ^4^ (mm)	4.54 (0.78)	4.79	4.55	4.54	4.30	4.67	4.42	0.02
MD_M_ ^5^ (mm)	27.1 (2.4)	26.8	18.5	27.1	18.9	22.6	23.0	0.20
RFI_M_ ^6^(g/day)	0 (183.1)	−3.2	−3.8	4.1	3.5	−3.5	3.8	0.81
		**Phase MIX**	**(N = 109 ^#^)**					
ADForI_M_ ^7^ (g/day)	1279 (224)	1276	1290	1268	1282	1283	1275	0.75
RForI_M_ ^8^(g/day)	0 (175.7)	−3.7	−3.0	3.3	4.0	−3.3	3.6	0.83

* X_C_ and X_M_ are phenotypes from the CONC and the MIX phase, respectively: ^1^ ADFI, average daily feed intake; ^2^ E-MW, end-phase metabolic weight; ^3^ ADG, average daily gain; ^4^ BFT, back fat thickness; ^5^ MD, back muscle depth; ^6^ RFI, residual feed intake; ^7^ ADForI, average daily forage intake; ^8^ RForI, residual forage intake. ^#^ Only animals from 2019 and 2020 had a precisely registered forage consumption; in 2018, forage and concentrate were distributed as a total mixed ration.

**Table 2 metabolites-12-00304-t002:** Number of lambs in each phenotypic class in the CONC phase according to their genetic line.

Phenotypic RFI_C_ (g/Day) ^1^					
**Genetic Line**		≤−75	]−75,0]	]0,75]	>75
	rfi−	62	38	17	18
	rfi+	20	19	40	63

^1^ The −75 and 75 thresholds were chosen because they were −0.5 SD and 0.5 SD of phenotypic RFI_C_, respectively.

**Table 3 metabolites-12-00304-t003:** Number of lambs in each phenotypic class in phase MIX according to their genetic line.

Phenotypic RFI_M_ ^1^ (g/Day)					
**Genetic Line**		≤−90	]−90,0]	]0,90]	>90
	rfi−	28	17	17	21
	rfi+	25	18	13	28

^1^ The −90 and 90 thresholds were chosen because they were −0.5 SD and 0.5 SD of RFI_M_, respectively.

## Data Availability

The datasets analyzed in the current study are available from the corresponding author upon reasonable request.

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
