# Peer review of "Evaluation of the Links between Lamb Feed Efficiency and Rumen and Plasma Metabolomic Data"

_metabolites, 2022, doi:10.3390/metabo12040304_

Round 1

Reviewer 1 Report

Revision has improved the manuscript, but English language is poor and needs to be improved.  Below I have commented on few language examples, but a thorough revision of language is needed.

Title, specially the term ‘Digging into’, is not very scientific and rather populistic, and I suggest to revise it, suggestion is ‘Evaluation of sheep feed efficiency using rumen and plasma metabolomics data’

Abstract: ‘Residual feed intake (RFI)’ is not a tool but rather a parameter. Therfore, change sentence to: ‘Residual feed intake (RFI) is a widely used parameter…

Line 14: replace ‘were led’ with ’were conducted’

Line 17: data = plural -> data were

Lines 19-21: Change: ‘Plasma citrate and malate were associated with genetically efficient animals while 19 amino acids such as L-threonine, L-serine and L-leucine as well as beta-hydroxyisovalerate were 20 associated with genetically inefficient animals under both diets’ to: ‘High levels of plasma citrate and malate were associated with genetically efficient animals while high levels of amino acids such as L-threonine, L-serine and L-leucine as well as beta-hydroxyisovalerate were  associated with genetically inefficient animals under both diets’

Lines 141-142: The sentence ‘Since concentrate intake was limited to 700 g/day in 2019 and 2020, total ration of  animals that ate less forage was somehow denser in energy and crude protein’ is unclear. The sentence reads as the animals were denser in energy and crude protein, which probably refers to the diet/feed and not the animals.

Line 156: ‘eight components’ refer to the PLS-DA model? This is unclear and sentence must be revised.

Lines 158-160: The sentence ‘The loadings of first component, by construction the most discriminant, highlighted the association between line rfi- and citrate and malate and between line rfi+ and L-leucine, L-threonine and beta-hydroxyisovalerate’ is poor and unclear.

Reviewer 2 Report

The paper examines impact of selection for feed efficiency on rumen and plasma metabolomic data. The paper is much improved. Some points for further improvement are outlined below 

L10: feed efficiency will not impact the cost of breeding??? it will impact the cost of production

L11: RFI is not a tool it is a trait? 

L12 & Throughout: metabolomic 

L35: conciliate. Not sure if this is the correct term

L40: What are the heritabilities for sheep

Results and Discussion 

Are the VFA results available? These should be included in the paper 

Methods 

What diet were the sires phenotypes on? This should be clear to the reader . Did you assess the change in ranking on the different diets? Or was it a separate group of animals used in the forage vs concentrate phases? The microbiome/metabolome could be selected for based on one diet but could re-rank based on a different diet?

In the mix phase was the time (Summer/fall) by rfi interaction tested?

Reviewer 3 Report

The present study has obtained a large amount of data. And they have also analyzed their data with a Omic idea. But I still have some comments here:

1) I would like to suggest the author to analyze the roles of differential metabolites using KEGG database. 

2) Why the author have not present the ruminal metabolites data, but just analyze the correlation between ruminal and plasma metabolites? 

Round 2

Reviewer 1 Report

The English language is improved, but the text is very lengthy with many redundant terms and phrases. It is suggested that the authors should reduce the discussion section by 10%. Below a few specific comments.

Line 412: delete ‘Finally’  (word is redundant and this is an odd way to start a new section)

Line 424: delete ‘In 2012’ (year is not essential for the text content)

Line 430: delete ‘in 2009’ (year is not essential for the text content)

Line 432: delete ‘in 2006’ (year is not essential for the text content)

Line 450: ‘notable’

Line 451: delete ’ in both phases’

Lines 459-460: Replace ‘Prior to the study, rumen metabolites were considered as potentially linked to feed  efficiency in two ways’ with ‘Rumen metabolites could potentially be linked to feed efficiency in two ways’

In several places the authors use the phrase ‘could stem from’. Meaning is unclear. Did the authors mean ‘originate from’? Please revise.

Author Response

This manuscript is a resubmission of an earlier submission. The following is a list of the peer review reports and author responses from that submission.

Round 1

Reviewer 1 Report

This paper assesses the differences in rumen and plasma metabolites in sheep divergent in feed efficiency. the experimental model is very difficult to follow and makes it impossible to assess the metabolomic data. If the animal model is not quantifying what is expected (RFI) the metabolomic data can not be assessed. See specific comments on the model below. 

Table 3: There a number of differences in the animal model based on the expectations of an RFI model. Metabolic BW and BFT should not be different if they were accounted for in the RFI model and suggest the populations are not actually divergent in RFI. It is also mentioned in line 101 that E-MW does not differ but is significant in Table 3. In the methods L585-622. It is unclear which animals from the concentrate diet are then fed the forage diet, when RFI is calculated and then from which animal samples are collected from. There is also mention of restriction of feed intake but no mention of timepoints L635. The methods need to be completely re-written to allow the reader to understand the animal populations 

Reviewer 2 Report

The manuscript by Touitou et al. presents a study where NMR-based metabolomics is applied on plasma and ruminal samples from sheep, and the overall aim is to elucidate the relationship between the derived metabolomes and feed efficiency. Apparently the authors compare two different genetic lines for this purpose. While this might be a sound approach, this strategy has to be more clear in text descriptions including abstract and results section. Overall, the applied methodologies and statistical analyses appear sound, however, I have concerns that authors have normalized spectral data including a residual water signal.

In general: metabolites should not be spelled with big capital. This should be revised throughout the manuscript.

Line 11: replace ’plasmatic’ with ’plasma’

Line 19: replace ‘No metabolite from rumen metabolomics analyses was discriminant..’ with ‘No metabolites from rumen metabolomics analyses were discriminant..’

Line 22: unclear what ‘lines’ refer to, please specify

Results:

I agree it is relevant to include information on metabolites detected, but the way that the authors lists the metabolites is rather tedious to read and it is recommended that information on missing metabolites is moved to supplementary material. Alternatively all this information should be moved to the materials and methods section.

The authors need to add more information on why they did as they did to make the text clear to the reader. When I read lines 146-155, I could not understand what classes that the PLS-DA model was supposed to discriminate? From figure 1, I guess it was the genotypes rfi -/rfi +? This should be clear from the text in the result section. In addition, the authors need to explain their approaches. What is AUROC and what is the impact of a value of 0.81? This needs to be explained. What is balanced error rate (BER) and how should it be interpreted? It is also difficult to understand the impact of 8 components in the model when is unclear how many variables and observations that was included in the model? How was missing metabolites handled in the PLS-DA model?

Figure 1: The PLS-DA scores plot does not really provide any information, why is this plot included? Since component 1 only explains 12% and in no way discriminate the two groups, would it not be better to use regression coefficients to elucidate important metabolites?

Tables should be provided in numerical order, and it therefore seems that table numbering should be revised.

Figure 2: The PLS-DA scores plot does not really provide any information, why is this plot included? Since component 1 only explains 9% and in no way discriminate the two groups, would it not be better to use regression coefficients to elucidate important metabolites?

Figures 6 and 7: Same questions as for Figure 1, 2.

Discussion:

In general the discussion is very long and it could be condensed without compromising the content.

Line 350: replace ‘article’ with ‘study’

Table 4 summarizing results from previous studies would be more appropriate for a review paper than an original research paper as presented here. I recommend that Table 4 is excluded.

Lines 426, 429 and elsewhere: replace ‘plasmatic’ with ‘plasma’

Line 544: delete ‘led’

Materials and methods:

Section 4.4.2: Why were data normalized BEFORE removing residual water signal? That means a poor water suppression in a spectrum will affect the quantification of metabolites?

Reviewer 3 Report

Comments to authors:

The manuscript by Touitou et al. deals with the metabolomic analysis by NMR of rumen and plasma samples from male lambs. Authors compare two different phase diets (concentrated and mixed) and want to establish a link between differentially expressed metabolites and feed efficiency. This paper might be of interest to readers in the breeder ecology and economy-related fields but, before I can recommend publication in Metabolites, there are several issues that must be addressed:

Main issues:

  • My main concern is the way authors select or highlight relevant metabolites. First, they rely on VIP values obtained from the PLS-DA models but then they switch on correlations (for example see Figures 2, 4, and 5). These correlations needs to be discussed with extreme care because it does not imply causation. For a more solid interpretation I strongly recommend authors to rely on statistically significant metabolites. This means that, in addition to the VIP, authors should include the corresponding p-values and box-plots for each metabolite. This will help visually interpreting the results.
  • I detected some inconsistencies and mistakes in the manuscript that made it somehow difficult to follow. Some specific comments are below:
    1. Introducing tables and figures should be done in order of appearance. For example, table 3A is the first one to be introduced (line 72) thus it should be named as Table 1. This is constantly repeated across the manuscript.
    2. Some Figures are not placed where they should be. Line 180: I believe authors mean Figure 2 (correlation matrix) and not Figure 1?
    3. Line 147: what do authors mean by Supplementary Data? Has this been uploaded?
    4. Figure 8: aspartate is highlighted as significant (blue color) however this metabolite was not even quantified in the plasma dataset? Can authors please double check?

Other comments:

Abstract is not so well formulated and authors should improve this. Some specific comments: on line 19 authors indicate no differences were found between lines, please include what you mean by lines here. Lines 19-23 include a very long statement and it is not so clear. Can authors please rephrase?

Line 92: this title is not complete. It should be something like “Growth phase under a mixed diet”.

Line 146: not clear what authors mean with fixed effects correction. Please, clarify.

Lines 148: can authors check what is the actual percentage explained by the different components? The first PC should always cover higher variability, not the second one… Check also Figure 1.

Lines 149-153: authors report that 7 metabolites had VIP higher than 1.5 in the first component but on Figure 1 the VIPs are calculated for the 8 components. This is a bit ambiguous and I would recommend authors to modify this. Moreover, why would authors want to include the 8 components if the first component is the one with the highest variability and the one in which the separation of the two groups take place?

Authors should better explain what is the different data used for Figure 1 and Figure S1. What does (-328,-75 and 75,546) even mean? Legend is not clear… I guess this is what is later on described on lines 703-706 but an explanation should have been given much earlier.

It should also be clearer why authors have to double check that differences are not found between the two different PLS-DA plots (again Fig 1 and Fig S1). What is the point that authors want to make? Please include in the manuscript.

Line 153: authors should describe what BER is and why this is useful. I think this is somehow described but much later on the manuscript (see lines 705-707).

Line 179: creatine and creatinine from Figure 1 have opposite directions according to the loading plots. Is this correlation negative then? Same applies for Choline and 2-oxoisovalerate.

Line 263: authors state that ANOVA p-value was not significant. Authors should include the p-value for each of the PLSDA models in the manuscript. Additionally to CV-ANOVA p-values, authors should include permutation tests for each PLS-DA plot that is in the manuscript. This will help to visualize whether models are overfitted.

Figure 7: there is a gold circle sample which is not filled near the center of the PLS-DA score plot. What does this mean?

Lines 650-651: How was the blood collected? What was the anticoagulant used? How was the samples stored after freezing? Many things are missing and must be specified.

Line 655: authors indicate that samples from 2018 and 2019 were prepared at the same time, and 2020 samples were prepared independently ( I assume at a much later stage). Did authors check whether there were batch effects present? Authors are strongly encouraged to plot the data as PCA score plot and color them according to time of preparation to see if there are specific trends based on the sample preparation time.

Line 677: authors discuss that the protein presence hampered utilization of TSP for quantification. Why didn’t authors precipitate the proteins in the samples with organic solvents, e.g. methanol? This is quite standard protocol in NMR-metabolomics studies.

Some typos I found:

-Line 58: highlighting is misspelled.

-Line 60: mentioned is also misspelled.

-Line 98: consumption is misspelled.

-Line 706: validation is misspelled.